# Alternate patterns of temperature variation bring about very different disease outcomes at different mean temperatures

**Charlotte Kunze[1,2†], Pepijn Luijckx[2]\*[†], Andrew L Jackson[2], Ian Donohue[2]**

[1]Institute for Chemistry and Biology of the Marine Environment [ICBM], Carl von Ossietzky University of Oldenburg, Oldenburg, Germany; [2]Department of Zoology, School of Natural Sciences, Trinity College Dublin, Dublin, Ireland

**Abstract** The dynamics of host-parasite interactions are highly temperature-dependent and may be modified by increasing frequency and intensity of climate-driven heat events. Here, we show that altered patterns of temperature variance lead to an almost order-of-magnitude shift in thermal performance of host and pathogen life-history traits over and above the effects of mean temperature and, moreover, that different temperature regimes affect these traits differently. We found that diurnal fluctuations of ±3°C lowered infection rates and reduced spore burden compared to constant temperatures in our focal host *Daphnia magna* exposed to the microsporidium parasite *Ordospora colligata*. In contrast, a 3-day heatwave (+6°C) did not affect infection rates, but increased spore burden (relative to constant temperatures with the same mean) at 16°C, while reducing burden at higher temperatures. We conclude that changing patterns of climate variation, superimposed on shifts in mean temperatures due to global warming, may have profound and unanticipated effects on disease dynamics.

**\*For correspondence:**
luijckxp@tcd.ie

[†]These authors contributed equally to this work

## Editor's evaluation

Kunze et al. demonstrate with a well-designed experiment that increases in mean temperature, and (extreme) variability in temperature regimes, have both consequences in host-pathogen interactions. The authors provide us with a realistic understanding of disease dynamics under climate change and identifies a need for mechanisms behind species interactions in fluctuating environments. This paper will be of interest to limnologists and disease ecologists, and it also provides a valuable information for epidemiologists.

## Introduction

One of the major challenges of the 21st century is understanding how infectious diseases, which have profound ecological and epidemiological impacts on human (*Hotez et al., 2014*), agricultural (*Chakraborty and Newton, 2011*), and wildlife (*Harvell et al., 2019*) populations, will be affected by climate change. It is now well-established that the interaction between hosts and their pathogens is sensitive to temperature (*Kirk et al., 2020*; *Rohr et al., 2013*). For example, disease transmission (*Ben-Horin et al., 2013*), host immunity (*Dittmar et al., 2014*; *Rohr and Raffel, 2010*), and pathogen growth (*Gehman et al., 2018*; *Kirk et al., 2018*) can increase with temperature, while other host-pathogen life-history traits such as lifespan and fecundity can decrease (*Altizer et al., 2013*). The interaction between temperature and multiple host and pathogen life-history traits

**eLife digest** Global warming is increasing average temperatures and causing extreme temperature fluctuations and heatwaves. These changes may affect when, where, and how often infectious disease outbreaks occur. This could have profound impacts on agriculture, human health, and wildlife.

Studying how extreme temperatures or temperature fluctuations alter infections in laboratory animals may help scientists to better understand the impact of climate change on disease. A small aquatic invertebrate, such as a water flea, is one good candidate for such studies. These tiny creatures can be grown in small glass jars in temperature-controlled aquariums.

Kunze, Luijckx et al. show that temperature fluctuations and heat waves have complex effects on parasitic infections in water fleas. In the experiments, water fleas housed with a parasite that infects them were exposed to constant temperatures, fluctuating temperatures, or three-day heatwaves, while being kept at a broad range of mean water temperatures. Then, Kunze, Luijckx et al. measured how these conditions affected the water fleas' longevity, reproduction, and parasite infections.

This revealed that temperature variations had a unique effect on the life span, and reproduction and infection rates of the water fleas, depending on the average water temperature the animals were kept at. Heatwaves drastically increased the number of parasites in the water fleas at an average water temperature of 16 °C but had no effect at all or decreased the number of parasites at 19 °C and 22 °C, respectively. Similarly, at high average water temperatures (>24 °C), temperature fluctuations reduced the number of water fleas infected with parasites and the number of parasites in each infected flea. Moreover, the maximum temperature at which parasites were able to cause infections was 5 °C lower under fluctuating temperatures than under constant temperatures.

Kunze and Luijckx et al. show that consistent high temperatures, temperature changes, extreme weather events, and mean water temperature affect disease outcomes in water fleas. More studies are needed to assess how temperature variations change the course of diseases in other organisms and to understand the underlying mechanisms. Learning more about disease-temperature interactions will help scientists predict climate change-driven disease outbreaks.

highlights the inherent complexity of temperature effects on infectious diseases. Indeed, each host or pathogen trait may have a unique dependency on temperature and it is their combined effect (i.e., $R_0$, disease outbreak, virulence) that is often of interest. However, while a growing body of theoretical (*Kirk et al., 2020*; *Rohr et al., 2013*) and empirical (*Ben-Horin et al., 2013*; *Dallas and Drake, 2016*; *Gehman et al., 2018*; *Kirk et al., 2020*; *Zhang et al., 2019*) studies have quantified the effect of rising mean temperatures on host and pathogen traits (such as, e.g., within-host growth [*Kirk et al., 2018*], pathogen transmission [*Kirk et al., 2019*], and epidemiology [*Gehman et al., 2018*; *Shocket et al., 2018*]), the influence of variable temperature regimes such as heatwaves and temperature fluctuations remains unresolved (*Claar and Wood, 2020*; *Rohr et al., 2013*).

Climate change is predicted to increase not only mean temperatures, but also temperature fluctuations and the frequency and intensity of extreme weather events (*Schär et al., 2004*; *Vasseur et al., 2014*). Such changes in temperature variance have the potential to modify host-pathogen dynamics (*Franke et al., 2019*; *Rohr et al., 2013*). For instance, diurnal temperature fluctuations have been shown to increase malaria transmission at the lower end of the thermal range (*Paaijmans et al., 2010*), while short-term temperature fluctuations led to reduced transmission success due to lower filtration rates in a *Daphnia*-pathogen system (*Dallas and Drake, 2016*). The effect of extreme heat events on host and pathogen traits is also highly variable and may depend on the magnitude, duration, and intensity of the applied heatwave (*Landis et al., 2012*; *Schreven et al., 2017*; *Zhang et al., 2019*). In a parasitoid-insect interaction, a heatwave of 5°C resulted in greater parasitoid development while a 10°C increase reduced parasitoid growth (*Schreven et al., 2017*). These apparent contrasting results in response to variation in temperature (here used to refer both to fluctuating temperature regimes and extreme heat events) imply that alternate temperature regimes or exposure to temperature shifts of different magnitudes will have distinct impacts on host-pathogen interactions. Indeed, whether all temperature variation acts in the same way or leads to different disease outcomes has been identified as a key open question in the field (*Rohr et al., 2013*).

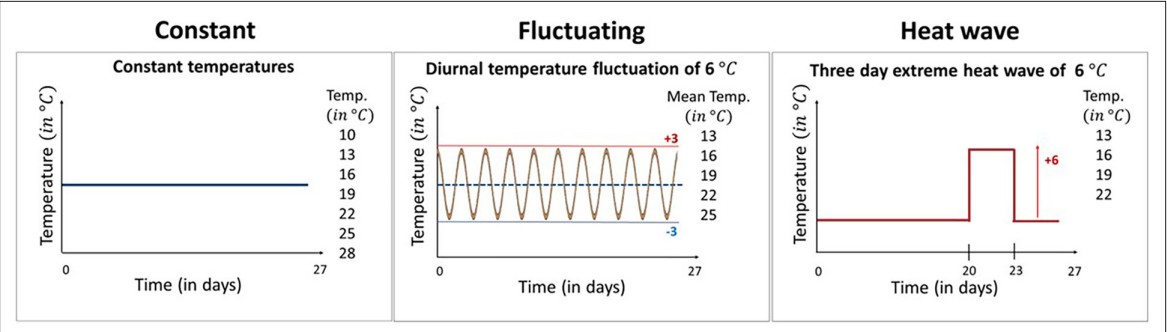

**Figure 1.** The three temperature regimes used in our experiment. Our experimental design comprised seven constant temperature regimes with temperatures ranging from 10°C to 28°C, five variable temperature regimes mimicking diurnal temperature fluctuations of ±3°C around the mean, and four heatwave regimes where temperatures were identical to the equivalent constant treatment except during a 3-day period between days 20 and 23 when temperatures were raised by 6°C. All temperature treatments were exposed to the *Daphnia* parasite *Ordospora* and to a placebo infection that served as a control for parasite exposure. Constant temperature regimes were replicated 12 times (7×12×2=168 individuals), while in the variable temperature regimes, the number of replicates was increased to 18 as we expected increased mortality in these treatments (5×18×2=180 and 4×18×2=144, respectively, for the fluctuating and heatwave regime). All animals were terminated after day 27 and fitness estimates were collected within 3 days.

Here, we examine the effect of different types of temperature variation on host-pathogen interactions across a broad range of mean temperatures. Specifically, we used the *Daphnia magna—Odospora colligata* host-pathogen system to test experimentally how temperature variation alters the thermal performance of both the host and the pathogen across their natural temperature range. *Daphnia* are a well-established ecological model system (*Miner et al., 2012*) used frequently in climate change studies (e.g., *Dallas and Drake, 2016*; *Hector et al., 2019*; *Kirk et al., 2020*), while *Ordospora* transmission is representative of a classical environmentally transmitted pathogen (i.e., it mimics diseases such as SARS-CoV-2 and *Vibrio cholerae*) and meets the assumptions of conventional epidemiological models (e.g., infection following mass action [*Kirk et al., 2019*], continuous shedding of infectious particles [*Ebert, 2005*] and little or no spatial structure within host populations). Our microcosm experiment comprised three distinct temperature regimes: constant temperatures and two variable temperature regimes with diurnal fluctuations of ±3°C and 3-day heatwaves of 6°C above ambient, all replicated over the natural temperature range of the model system (i.e., 10–28°C, *Figure 1*). These variable temperature regimes were selected to mimic naturally occurring temperature events in habitats our study organisms encounter naturally (i.e., small ponds and rock pools) (*Jacobs et al., 2008*; *Kuha et al., 2018*).

During the experiment, we measured host longevity, fecundity, infection status, and the number of *O. colligate* spores within the host gut (see Materials and methods for details). All measurements were conducted on individually kept *Daphnia* with up to 18 replicates per measurement. To compare the three different temperature regimes (i.e., constant, diurnal fluctuations, and heatwave; *Figure 1*), we fitted a Beta Function using a Bayesian framework. While there are numerous non-linear functions that can be used to fit thermal performance curves (e.g., Briére, Ratkowsky equation), the advantage of using the Beta Function is that it provides realistic predictions when extrapolating beyond the measured thermal range and that each of its parameters has a clear a biological meaning (see *Shi et al., 2016*, for a comparison of thermal performance equations), where $F_m$ is the fitness at optimal performance for the fitted host or parasite trait, $T_{opt}$ is the temperature at optimal performance, and $T_{min}$ and $T_{max}$ are, respectively, the critical minimum and maximum temperatures over which fitness of the trait becomes unviable.

## Results

Diurnal temperature fluctuations narrowed the thermal performance curve for infectivity compared with constant temperatures (*Figure 2A*). The estimated (using the Beta Function) maximum temperature at which spores were able to cause infections was 5°C lower under fluctuating temperatures than under constant temperatures (*Figure 2A*; $T_{max}$=25°C for fluctuating vs. 30°C for constant; confidence

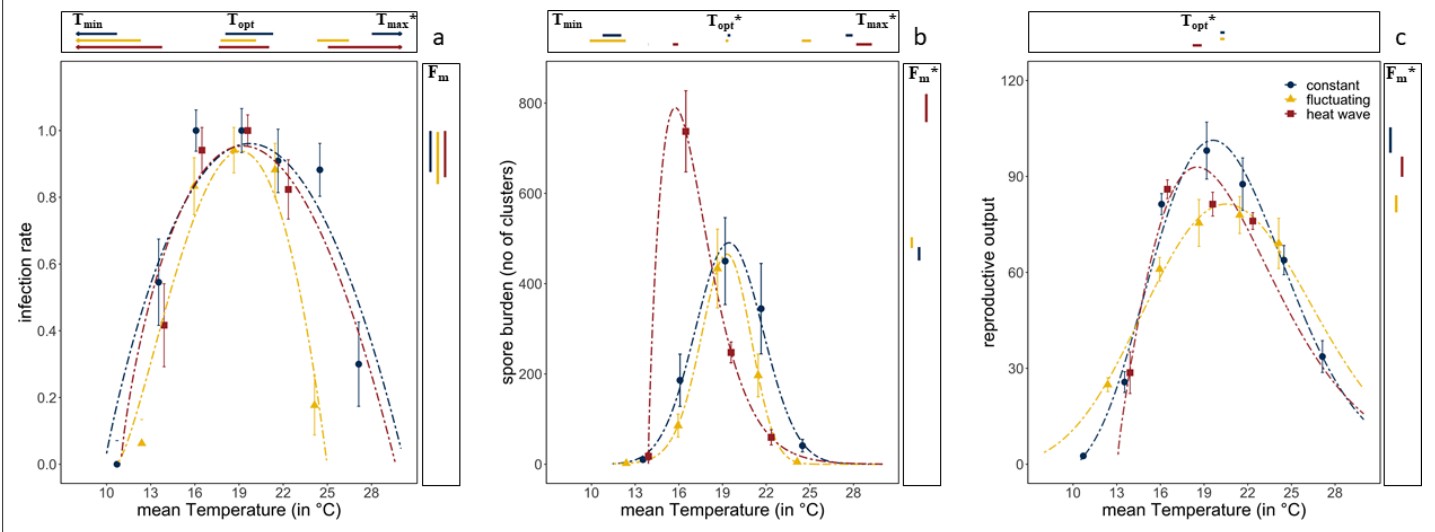

**Figure 2.** Thermal performance curves of host and parasite life-history traits across our three temperature regimes. (**a**) Infection rates of *Ordospora* in its *Daphnia* host. (**b**) Mean number of spore clusters in infected *Daphnia* at the end of the experiment. (**c**) Reproductive output of the host when exposed to *Ordospora* (for a comparison of individuals exposed to *Ordospora* and the controls that were exposed to a placebo, see ***Figure 3***). For all panels, the constant temperature regime is in blue, the diurnally fluctuating regime in yellow, and the heatwave in red. Points present the observed mean values for the measured traits and dashed lines provide the fit for the Beta Function. 95% confidence intervals of Beta Function estimates for minimum, optimal, and maximum temperature (respectively, $T_{min}/T_{opt}/T_{max}$) are shown above the x-axis. The estimate for the optimal value of the life-history trait ($F_m$) and its 95% confidence interval is displayed to the right of each panel. Significant differences (non-overlapping 95% confidence intervals) in parameter estimates are highlighted with an asterisk. Error bars on data points indicate standard error. Beta Function parameter estimates displayed in this figure can be found in ***Supplementary files 2–4***.

intervals for $T_{max}$ do not overlap). The thermal performance curve for infectivity under the heatwave, where temperatures were raised by 6°C for 3 days and then returned to constant temperature (***Figure 1***), was almost identical to that under constant temperature (all confidence intervals overlap, ***Figure 2*** and ***Supplementary file 2***). However, unlike the constant temperature regime, the heatwave did not differ from the fluctuating regime, as estimates for the maximum temperature had broad confidence intervals, likely caused by lack of data at the higher temperatures. Remaining parameter estimates of the Beta Function were similar for the three temperature regimes, with the highest rate of infection at 19°C, a maximum infection rate of ~95% infection and no infections under 10°C (***Figure 2A*** and ***Supplementary file 2***; confidence intervals overlap for $T_{opt}$, $F_m$, and $T_{min}$). Thus, while diurnal fluctuations led to less infection at higher temperatures, a heatwave did not alter infection rates.

Spore burden of the two variable temperature regimes deviated from both the constant temperature regime and from each other (***Figure 2B***). Consistent with infection rates, daily temperature fluctuation led to a lower maximum temperature (by ~3°C) for parasite growth within the host, resulting in a narrowed thermal performance curve for burden compared with the other temperature regimes (***Figure 2B***, ***Supplementary file 3***, non-overlapping confidence intervals for $T_{max}$). This is supported further by the consistently lower spore burden for the fluctuating regime when compared with the constant temperature regime except near the optimum temperature of 19°C, where spore burdens of both temperature regimes were similar (confidence intervals for $T_{opt}$ and $F_m$ overlap). While infection rates and burden showed a similar thermal performance for diurnal fluctuations (both narrowing), the response to the heatwave differed between infection and burden (***Figure 2***). Compared to the constant temperature regime, spore burden in the heatwave showed a shift in the optimum temperature (from 19.4°C to 15.7°C), and an increase in the number of spore clusters (***Figure 2B***, confidence intervals for $T_{opt}$ and $F_m$ do not overlap). However, while spore burden was different at ~16°C, spore burden at ~19°C was nearly identical for all three temperature regimes. Moreover, due to the opposite effects at 16°C for both variable temperature regimes (i.e., a narrowing of performance under fluctuating temperatures, exacerbation under heatwave), spore burden at this temperature differed by almost an order of magnitude (i.e., 86 vs. 737 spore clusters).

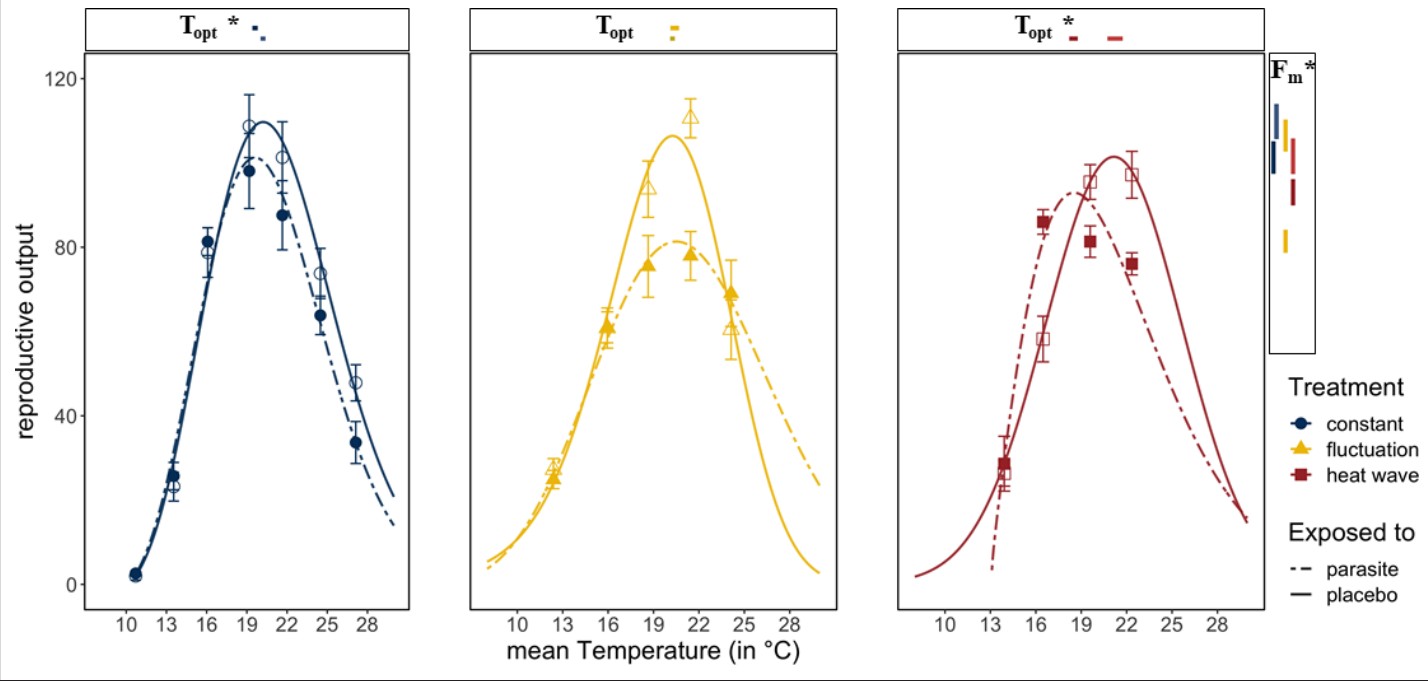

**Figure 3.** Reproductive success in *Daphnia* exposed to *Ordospora* and controls that were exposed to a placebo. *Daphnia* exposed to the parasite (dotted lines and filled symbols) produce less offspring than control individuals (solid lines and open symbols). Lines are the fitted Beta Functions for the different temperature regimes (constant temperature regime in blue, the diurnally fluctuating regime in yellow, and the heatwave in red). 95% confidence intervals of maximum reproductive output ($F_m$) are shown to the right, and the temperature of this optimum ($T_{opt}$) is shown above the x-axis. Significant differences in parameter estimates of the Beta Function are highlighted with an asterisk. Estimates for minimum and maximum temperatures are not displayed as we used restrictive priors. Error bars on data points indicate standard error. Beta Function parameter estimates displayed in this figure can be found in ***Supplementary file 4***.

Host fitness was generally reduced when exposed to *Ordospora* spores or when experiencing variable temperature regimes. *Daphnia* exposed to the parasite had lower reproductive success near the optimum temperature (~20°C) compared to control animals that were not exposed to *Ordospora* (non-overlapping confidence intervals for $F_m$) and lost between 8% (constant) and 24% (diurnal fluctuation) of reproductive output (***Figure 3***). Comparing host performance among the different temperature regimes shows that animals exposed to *Ordospora* in variable temperatures had lower reproductive success (***Figure 2C***, ***Supplementary file 4***, non-overlapping 95% confidence intervals for $F_m$), with a small shift (1.1°C) in their thermal optimum under the heatwave regime (***Figure 2C***, non-overlapping 95% confidence intervals for $T_{opt}$). Control animals that were not exposed to *Ordospora* also had lower fitness after the heatwave (***Figure 3***, non-overlapping 95% confidence intervals for $F_m$) and, while reproduction at the optimal temperature of the control animals experiencing diurnal fluctuations was lower, confidence intervals overlapped with the constant temperature regime (***Figure 3***). The host response to the variable temperature regimes differed from that of the pathogen (compare thermal performance curves for the heatwave and diurnal fluctuating regimes between ***Figure 2A and B & C***). While host performance was reduced (lower $F_m$) under both variable temperature regimes, parasite traits showed either a narrowing of the performance curve (for diurnal fluctuations) or no effect and greatly increased performance (for infection and burden under the heatwave).

## Discussion

We show that, not only does temperature variation alter the thermal performance of host and pathogen life-history traits in a unique way—driving a shift in performance up to order-of-magnitude over and above the effect of mean temperature—but that the type of variation and the mean temperature at which it occurs are also critical. Indeed, each of the life-history traits we measured was affected differently by thermal variation. With global warming altering the mean and variance of temperature

around the world, how this affects diseases and their dynamics is a critical outstanding question (*Claar and Wood, 2020*; *Rohr et al., 2013*). Our results demonstrate that the combined effect of changing temperature mean and variance can be highly complex, and may alter the vulnerability of host populations (*Harvell et al., 2019*), affect the evolution of host and parasites (*Buckley and Huey, 2016*), and, therefore, impede our ability to accurately predict future disease outbreaks.

Infection rates were reduced at higher temperatures when animals experienced diurnal fluctuations but not after experiencing a heatwave. Our estimates of maximum temperature for the heatwave, however, have broad confidence intervals, likely due to the lack of data at high temperatures, and extrapolations of our results beyond our highest heatwave temperature (which was 22°C and reached 28°C during the 3-day heatwave which was close to the estimated maximum thermal tolerance of the parasite) should be interpreted with caution. In *Daphnia*, filtration rates determine the contact rate between host and pathogen, and a reduction in filtration can thus lead to reduced levels of infection (*Hall et al., 2010*). As filtration rates of *D. magna* decline at higher temperatures (*Kirk et al., 2019*), average infection rates under diurnal temperature fluctuations would also be expected to be lower due to the non-linear nature of the thermal performance curve (i.e., Jensen's inequality; *Dowd et al., 2015*). In addition, infection probability in our study system decreases sharply when temperatures surpass 22°C (*Kirk et al., 2019*), reducing infection rates under fluctuating temperatures that exceed this temperature (again, due to Jensen's inequality). In systems where immune function depends on temperature (e.g., insects, mosquitos, and ectotherms in general; *Paaijmans et al., 2013*), heatwaves may interact with the immune system in complex ways (*Murdock et al., 2012*), particularly when the heatwave occurs early in the infection process. However, given that our heatwave occurred 20 days post-infection and that *Daphnia* are not known to recover from infection (*Ebert, 2005*), the effect of the heatwave on established infections may have been limited. Absence of an effect of a heatwave on infection rates has also been found for a pipefish-trematode host-parasite system (*Landis et al., 2012*). However, though the heatwave did not affect infection rates in our experiment, it did affect parasite burden.

Our results show that different types of temperature variation can alter parasite burden and thus affect pathogen growth within the host. While diurnal temperature fluctuations and heatwaves brought about an almost order of magnitude difference in spore burden at a mean temperature of 16°C, no differences were observed at ~19°C. Generally, similar to infection rates, the thermal performance curve for spore burden narrowed under fluctuating temperatures, as predicted by averaging over the non-linear thermal performance curve (*Denny, 2017*; *Dowd et al., 2015*). The impact of diurnal temperature fluctuations on parasite fitness has been studied previously, with multiple studies suggesting a shift in the thermal performance of parasite fitness under fluctuating temperatures (*Dallas and Drake, 2016*; *Duncan et al., 2011*; *Greenspan et al., 2017a*; *Paaijmans et al., 2010*). Indeed, our findings that *Ordospora* has a narrower thermal performance for spore burden and infectivity under fluctuating temperatures adds to a growing body of evidence (*Dallas and Drake, 2016*; *Greenspan et al., 2017b*; *Hector et al., 2019*; *Roth et al., 2010*) suggesting that estimates and predictions that ignore temperature variation may over- or underestimate disease burden and prevalence (*Greenspan et al., 2017a*; *Raffel et al., 2012*; *Rohr et al., 2013*). Moreover, with almost an order-of-magnitude difference between both our two variable temperature regimes at some, though not all, temperatures, our results highlight that both the context and type of temperature variance need to be considered when trying to understand how pathogen performance may be affected by climate change.

Spore burden increased following heatwaves, but the effect depended on the mean temperature to which the heatwave was applied. Indeed, the heatwave had either higher, similar or lower spore burden compared to the equivalent constant temperature regime. It was shown recently in a fish-tapeworm host-parasite system that parasite growth, egg production, and the number of first-stage larvae increased after a 1-week exposure to higher temperatures (increase up to 7.5°C) (*Franke et al., 2019*). Our findings corroborate that heatwaves associated with climate change may, under some conditions, increase disease burden. Indeed, we found a considerable increase in spore burden and a shift in the optimum temperature following a 3-day increase in temperature of 6°C at 16°C. Although some studies have reported increased disease susceptibility following heatwaves (*Dittmar et al., 2014*; *Roth et al., 2010*), others found no effect on immune function (*Stahlschmidt et al., 2017*) or reduced disease performance after exposure to high temperatures (*Fayer et al., 1998*). Our results

may explain these conflicting findings—we found that the effects of a heatwave on spore burden are contingent on the mean temperature to which the heatwave is applied. That is, our results show that the heatwave has either lower or higher burden than equivalent constant temperatures. This context-dependency of heatwaves is supported further by studies in both plant-endoparasite (*Schreven et al., 2017*) and herbivore-parasitoid (*Zhang et al., 2019*) systems, which showed that the effect of a heatwave on parasite traits depended on the amplitude of the extreme event. As highlighted by a recent review (*Claar and Wood, 2020*), effects of warming events on disease traits remain difficult to generalize, and more studies and insight into underlying principles and mechanisms are needed to forecast the effect of extreme heat events on disease dynamics. Indeed, while it is clear from our experiment that a short, 3-day increase in temperature can drastically alter the thermal performance curve for parasite burden, the exact mechanism(s) underlying this change remains unidentified.

Differences in acclimatization speeds between hosts and pathogens may explain the observed increase in burden of *Ordospora* at 16°C following a heatwave. According to the temperature variability hypothesis (*Raffel et al., 2012*; *Rohr et al., 2013*), parasites, which have faster metabolic rates due to their smaller size, should acclimatise more rapidly to changing temperatures than their larger hosts. In unpredictable variable environments, such as our heatwave regime, parasites should thus have an advantage over their hosts. Moreover, host resistance may also decrease as a result of the trade-off between the energy demand for acclimatization and immunity (*Nelson and Demas, 1996*). That varying temperature can lead to higher infection prevalence has been established in Cuban tree frogs, red-spotted newts, and abalone (*Ben-Horin et al., 2013*; *Raffel et al., 2012*). While this hypothesis may explain our observation of high burden for the heatwave near 16°C, it does not, however, explain why the response depends on the mean (i.e., lower performance at higher temperatures). Though *Ordospora* should have an overall advantage under the temperature variability hypothesis, the realized advantage may be smaller as its thermal range is more restricted than its host (*Kirk et al., 2018*). The heatwave may thus cause proportionally more stress in the parasite than the host at high temperatures, consistent with the thermal stress hypothesis, which suggests that a shift in temperature may reduce the performance of either host or parasite (*Paull et al., 2015*). Indeed, that thermal stress can affect host and pathogen performance has been well supported (*Gehman et al., 2018*; *Kirk et al., 2019*; *Schreven et al., 2017*; *Zhang et al., 2019*). Alternatively, the observed increase in parasite burden due to heatwaves may be system-specific and not explained by differences in acclimatization speed. Estimates show that growth rates of *Ordospora* increase by a factor of 5 between 20°C and 24°C before declining again (*Kirk et al., 2018*). While the optimal performance of *Ordospora* occurs around 19°C, due to the balance of thermal performance curves of other host and pathogen traits (e.g., mortality, infectivity, etc.), a temporary increase to 22°C, as occurred under our heatwave at 16°C, may thus have exacerbated pathogen growth, particularly if different traits react differently to a temperature disturbance, which may have disrupted the balance between host and parasite.

Changes in host fecundity in response to temperature variation differed to the response of both parasite traits (i.e., infectivity and spore burden) we measured. While infectivity and burden had either a narrower thermal performance curve or showed a heightened and shifted peak, temperature variation lowered the reproductive output of the host near the thermal optimum. A reduction in reproductive output of the host under variable temperatures is consistent with previous work both on *Daphnia* (*Schwartz et al., 2016*) and in other systems (*Craig and Kipling, 1983*; *Uvarov et al., 2011*). Similarly, a reduction in host fecundity due to parasitism is well established (*Ebert, 2005*). Infection may also reduce the thermal tolerance of the host (*Hector et al., 2019*), which would explain the small shift of the thermal optimum for host reproduction under the heatwave regime. While host responses are thus consistent with expectations, the distinct responses to the different temperature regimes of the different life-history traits we measured (i.e., host fecundity, parasite infectively, and parasite burden) highlight that the effects of temperature variation on host-pathogen systems are complex. When trying to model disease dynamics and outbreaks, we often include a multitude of host and pathogen traits, each with their own thermal dependencies. Recent studies have made advances in predicting disease growth and spread under rising mean temperatures, integrating approaches, and identifying mechanisms that can capture and predict the thermal performance of host and pathogen traits within epidemiological models (e.g., metabolic theory) (*Kirk et al., 2020*). It remains to be seen, however, whether such modeling frameworks can be extended to incorporate temperature variation,

especially considering the distinct responses for the life-history traits we measured to each of our variable temperature regimes.

Our study shows that temperature variation alters the outcome of host-pathogen interactions in complex ways. Not only does temperature variation affect different host and pathogen life-history traits in a distinct way, but the type of variation and the mean temperature to which it is applied also matters, with up to an order of magnitude change between diurnal fluctuations in temperature and extreme heat events. With global warming altering both the mean and variance of temperature around the world, we can thus expect to see unanticipated changes in disease dynamics of host-pathogen systems. Indeed, extreme temperature events such as *El Niño* have been linked to disease-driven collapses of keystone predators (*Harvell et al., 2019*), increases in diseases such as dengue and cholera (*Anyamba et al., 2019*), and shifts in the geographic distribution of pathogens (*Claar and Wood, 2020*). While temperature variation can thus affect disease dynamics in human, wildlife, and livestock populations—with potentially devastating economic and health consequences (*Altizer et al., 2013*)—the complexity of the effects of temperature and its variation currently limits our ability to move beyond system-specific predictions, in particular for extreme temperature events (*Claar and Wood, 2020*). We conclude that improving our mechanistic understanding of the role of temperature variation on disease dynamics, and exploring the generality of its effects and how it affects thermal performance curves of both hosts and parasites (*Claar and Wood, 2020*), are critical to predicting disease dynamics in a warming world.

## Materials and methods

### Study system

The crustacean *D. magna* plays a key role in ecosystem functioning. *Daphnia* are filter feeders that consume planktonic algae and other microorganisms, thus promoting water transparency and helping to prevent algal blooms (*Miner et al., 2012*). They are a key food source for planktivorous fish, constitute a major part in the food web (*Ebert, 2005*), and play a key role in nutrient cycling (*Elser et al., 2000*). Across its range, *Daphnia* is affected by a broad variety of pathogens. Here, we use *O. colligata*, a widely distributed microsporidium parasite that is only known to infect *D. magna*. This gut parasite has been recently used as a model to understand how changes in mean temperatures under global warming may affect host-parasite systems (*Kirk et al., 2020*). However, the effects of temperature variance remain unstudied. *Daphnia* become infected when they accidentally ingest water borne spores of *Ordospora* while filter feeding. After successful establishment, spores divide intracellularly in the gut epithelium of *D. magna* (*Larsson et al., 1997*) until they form a cluster of 32–64 spores. Spores are then released either to the environment or go on to infecting neighboring cells after *O. colligata* lyses the cell.

### Experimental set-up

In the laboratory, we established water baths with temperatures ranging from 10°C to 28°C. Each bath was regulated with a temperature controller (Inkbird ITC-308) that interfaced with cooling (Hailea HC300A) and heating (EHEIM JÄGER 300 W) units. Pumps (Micro-Jet Oxy) were used to create constant flow, which ensured equal temperature distribution within the water baths. Each bath held up to 99 microcosms and was kept under natural lighting conditions (16:8 light:dark). Temperature and light intensity were recorded using HOBO loggers housed in spare microcosms. Each microcosm was filled with up to 80 ml of Artificial *Daphnia* Medium (ADaM, modified to use only 5% of the recommended selenium dioxide concentration; *Klüttgen et al., 1994*).

To test for the effect of changing both mean temperature and patterns of temperature variation in our host-parasite system, we created three different temperature regimes: one constant and two variable temperature regimes, the latter comprising diurnal temperature fluctuations and a heatwave (*Figure 1*). In the constant temperature regime, individual *Daphnia* were kept at one of seven temperatures for the whole experimental period (i.e., 10, 13, 16, 19, 22, 25, and 28°C). The diurnal fluctuation regime comprised five temperature levels, which experienced the same mean temperature as the constant regimes but with a fluctuation of ±3°C every 12 hr (i.e., 10–16°C, 13–19°C, 16–22°C, 19–25°C, and 22–28°C), mimicking diurnal fluctuations in small rock pools (*Jacobs et al., 2008*). The heatwave was performed at four different temperature levels (13, 16, 19, and 22°C), with conditions

identical to the constant regime except for an increase of 6°C for 72 hr, 20 days after animals were exposed to the parasite, mimicking a short heatwave (**Kuha et al., 2018**). We chose these temperature levels because of their relevance for our host and pathogen system, as no infection occurs below 12°C and hosts have high mortality above 30°C (**Kirk et al., 2018**; **Kirk et al., 2019**). Animals were kept individually in microcosms, organized into trays, and repositioned daily to avoid positioning effects. In each temperature regime, half of the microcosms were exposed to the parasite while the other half served as controls. For each of the constant temperature levels, we used 12 replicates for both animals exposed to *Ordospora* and control animals that received a placebo exposure. However, as we expected greater mortality in the variable temperature regimes (**Régnière et al., 2012**), we increased the number of replicates of these regimes to 18. We based this number of replicates on experience with previous temperature experiments with the *Daphnia-Ordospora* system (**Kirk et al., 2019**).

The *Daphnia* genotype (clone FI-OER-3-3), we used was previously isolated from a rock pool at Tvärminne archipelago, Finland and propagated clonally in the laboratory. To generate sufficient animals for the experiment, we grew *Daphnia* asexually under standardized conditions for 3 weeks. Animals were raised in small populations (20 400 ml microcosms, 12 animals per microcosm) under continuous light at 20°C. The medium (ADaM) was replaced at least twice a week and *Daphnia* were fed ad libitum with *Scenedesmus* algae (*Scenedesmus* sp.), which was grown in batch cultures at 20°C in WC Medium (**Kilham et al., 1998**) under nutrient- and light-saturated conditions. The experiment was initiated by collecting a cohort of female juveniles (~600 females up to 72 hr old) from the small population microcosms. Individual juveniles were then randomly transferred into 100 ml glass microcosms filled with 40 ml ADaM. These glass microcosms were placed into their assigned water baths and, after an acclimation period of 24 hr, the animals were exposed to the parasite by adding 1 ml medium containing ~10,000 spores of *O. colligata*. This spore solution was prepared by crushing 3560 infected *D. magna* individuals with known average burden (determined by using phase-contrast microscopy on a sub-sample), using mortar and pestle and diluting down the resulting spore slurry. The unexposed controls received a placebo exposure consisting of crushed uninfected animals diluted in medium. Animals were exposed either to the parasite or placebo for 6 days and were transferred subsequently to clean microcosms with fresh medium (80 ml of ADaM) twice a week until the end of the experiment. Animals were fed four times a week with an increasing amount of algae to accommodate the increased food demand of the growing animals (from 4 million algae ml$^{-1}$ at the start of the experiment to 10 million algae ml$^{-1}$ by day 10, which was maintained until the end of the experiment). Between transfers, evaporation of the medium was offset by refilling microcosms daily with 50-50 ADaM-distilled water.

## Measurements of host and parasite life-history traits

To obtain fitness estimates for the host, we counted the offspring produced and checked mortality of all animals daily. Infection status and spore burden (i.e., the number of spores inside the host) were assessed upon death by dissecting individuals and counting the number of spore clusters (each cluster holds up to 64 parasite spores) in the gut with phase-contrast microscopy (400× magnification). Any animals that remained alive until the end of the experiment (day 27) were terminated within 3 days, dissected and their infection status and burden determined without the observer being aware of the identity of the sample. Because infections cannot be diagnosed accurately in early infection stages, animals that died before day 11 were not considered in analyses. Any male *Daphnia* that were misidentified as female at the start of the experiment were also excluded. In addition, to prevent potentially confounding effects of animals that died early (where the parasite had less time to grow) as having lower spore burden, we included only animals from the last day of the experiment in the analysis of spore burden. Note that, to facilitate good estimates of spore burden, we terminated most hosts before natural death occurred, which limits our ability to assess the effects of virulence (host mortality, reduced fecundity).

## Data analyses

Analyses were performed using R version 3.6.1 (**R Development Core Team, 2018**) interfacing with JAGS (**Lunn et al., 2009**; **Plummer et al., 2006**). Datafiles and code are available at https://github.com/charlyknz/HostParasite (**Kunze, 2022**; copy archived at swh:1:rev:5f2604fe866f547dd80d5a77

f99ef8887b9f10e1). A Beta Function was fitted to each of our different fitness estimates (i.e., host fecundity, parasite infectivity, and burden) for each of the three temperature regimes, as:

$$f = F_m \left( \frac{T_{max} - T}{T_{max} - T_{opt}} \right) \left( \frac{T - T_{min}}{T_{opt} - T_{min}} \right)^{\left( \frac{T_{opt} - T_{min}}{T_{max} - T_{opt}} \right)}$$

where $f$ is fitness at temperature $T$, $F_m$ is estimated fitness at optimal performance for the fitted host or parasite trait, $T_{opt}$ is temperature at optimal performance, and $T_{min}$ and $T_{max}$ are, respectively, the critical minimum and maximum temperatures over which fitness of the trait becomes unviable. This non-linear function has been shown to capture thermal performance accurately (*Niehaus et al., 2012*) and has the advantage that all four parameters in the equation have clear biological meaning.

To determine the effect of both mean and variation in temperature on host and pathogen traits, we used a Poisson distribution for reproductive output (number of offspring per individual) and spore burden (number of spore clusters produced by the parasite). For pathogen infectivity, we used a binomial distribution. Models were fitted using the MCMC fitting algorithm called from R. All models were fitted in a Bayesian framework with JAGS (*Lunn et al., 2009*; *Plummer et al., 2006*), while allowing for separate parameter values for each of the different temperature regimes. Priors for temperature effects were specified in order to satisfy the necessary condition $T_{min} \leq T_{opt} \leq T_{max}$ and informed by previous work (see *Supplementary file 1* for the priors) (*Kirk et al., 2018*; *Kirk et al., 2019*; *Kirk et al., 2020*). The posterior distribution of all parameters was estimated using three chains, 10,000 posterior draws which were then thinned by 5 to yield 6000 samples (3*10,000/5). Model convergence was checked using the Gelman-Rubin diagnostic.

## Acknowledgements

The authors thank Dieter Ebert and Jürgen Hottinger for provision of the biological materials, Alison Boyce for technical assistance in creating the water baths and Maren Striebel for helpful comments on the manuscript. ALJ was funded by an Irish Research Council grant IRCLA/2017/186. PL was funded by a Science Foundation Ireland Frontiers for the Future grant 19/FFP/6839.

## Additional information

### Funding

| Funder | Grant reference number | Author |
|---|---|---|
| Science Foundation Ireland | 19/FFP/6839 | Pepijn Luijckx |
| Irish Research Council | IRCLA/2017/186 | Andrew L Jackson |

The funders had no role in study design, data collection and interpretation, or the decision to submit the work for publication.

### Author contributions

Charlotte Kunze, Conceptualization, Formal analysis, Investigation, Methodology, Writing – original draft, Writing – review and editing; Pepijn Luijckx, Conceptualization, Funding acquisition, Investigation, Methodology, Resources, Supervision, Writing – original draft, Writing – review and editing; Andrew L Jackson, Formal analysis, Writing – review and editing; Ian Donohue, Investigation, Supervision, Writing – review and editing

### Author ORCIDs

Charlotte Kunze ⓘ http://orcid.org/0000-0002-1130-7417
Pepijn Luijckx ⓘ http://orcid.org/0000-0002-8173-6727
Ian Donohue ⓘ http://orcid.org/0000-0002-4698-6448

### Decision letter and Author response

Decision letter https://doi.org/10.7554/eLife.72861.sa1
Author response https://doi.org/10.7554/eLife.72861.sa2

## Additional files

### Supplementary files

- Transparent reporting form
- Supplementary file 1. Priors for the parameters in the Beta Function.
- Supplementary file 2. Parameter estimates of the Beta Function for infection rates.
- Supplementary file 3. Parameter estimates of the Beta Function for spore burden of *Ordospora colligata*.
- Supplementary file 4. Parameter estimates of the Beta Function for reproductive success of *Daphnia magna*.

### Data availability

Data and code are available on GitHub https://github.com/charlyknz/HostParasite.git (copy archived at swh:1:rev:5f2604fe866f547dd80d5a77f99ef8887b9f10e1).

The following dataset was generated:

| Author(s) | Year | Dataset title | Dataset URL | Database and Identifier |
|---|---|---|---|---|
| Kunze C, Luijckx P, Jackson A, Donohui I | 2021 | Alternate patterns of temperature variation bring about very different disease outcomes at different mean temperatures | https://doi.org/ 10.5061/dryad. w0vt4b8s3 | Dryad Digital Repository, 10.5061/dryad.w0vt4b8s3 |

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
