## [Editor Report]

Kunze et al. demonstrate with a well-designed experiment that increases in mean temperature, and (extreme) variability in temperature regimes, have both consequences in host-pathogen interactions. The authors provide us with a realistic understanding of disease dynamics under climate change and identifies a need for mechanisms behind species interactions in fluctuating environments. This paper will be of interest to limnologists and disease ecologists, and it also provides a valuable information for epidemiologists.

---

## [Decision Letter]

**Decision letter after peer review:**

Thank you for submitting your article "Alternate patterns of temperature variation bring about very different disease outcomes at different mean temperatures" for consideration by *eLife*. Your article has been reviewed by 2 peer reviewers, one of whom is a member of our Board of Reviewing Editors, and the evaluation has been overseen by Dominique Soldati-Favre as the Senior Editor. The reviewers have opted to remain anonymous.

Essential revisions:

Both reviewers have identified a set of issues that need to be solved before a final decision is taken.

*Reviewer #1 (Recommendations for the authors):*

1. Line 84. Please mention the advantage of a β function over alternatives.

2. Line 105. 30C? Not sure I understood correctly, but max temp in constant regime in Figure 1 is set in 28C?

3. Please make it easy to understand the word *exposure*… across the ms you expose Daphnia to different temps, but also to pathogens, and unexposed controls were exposed to a Placebo. Figure 3 legend should be more specific, if you want to expose the reader to some clarity!

*Reviewer #2 (Recommendations for the authors):*

As I mentioned I have no major concerns about the manuscript, I liked it very much.

My points (or advices) are as follows:

– The first (lines 184-187) and the third (lines 188-190) sentences of the discussion basically deliver the same information. I'd suggest to remove the latter.

– lines 196-197: "Infection rates were reduced at higher temperatures when animals experienced diurnal fluctuations but not after experiencing a heatwave." – given that (as authors suggest in the Results section – lines 108-111) the authors lack the data over 22C for heat wave treatment (for absolutely justified reasons), the statement that infection rates were not reduced at higher temperatures in this treatment has a limited support. It is always risky to draw conclusions from data extrapolated beyond the last measurement point. Therefore, I think it would be good to add to the above mentioned statement a similar in character 'warning' as authors did in the Results section.

---

## [Author Response]

Reviewer #1 (Recommendations for the authors):1. Line 84. Please mention the advantage of a β function over alternatives.

Thank you for this suggestion. We have expanded on the advantages of the β function over some of its alternatives and included a reference which compares multiple thermal performance curves. Accordingly, we have modified this section as follows (lines 83-87 on our revised manuscript):

“While there are numerous non-linear functions that can be used to fit thermal performance curves (e.g., Briére, Ratkowsky equation) the advantage of using the Β Function is that it provides realistic predictions when extrapolating beyond the measured thermal range and that each of its parameters has a clear a biological meaning (see Shi et al., 2016, for a comparison of thermal performance equations)

2. Line 105. 30C? Not sure I understood correctly, but max temp in constant regime in Figure 1 is set in 28C?

Thank you for pointing out that this was unclear. The 30℃ refers to the estimate generated by the Β Function, and not to the observations taken in this study. While this estimate for the maximum temperature is 2 ℃ beyond our last measured datapoint, its confidence intervals (28-34 ℃) and mean (30 ℃) fall well within the expectations based on previous studies (Kirk et al., 2018 Plos Biology and Kirk et al., 2019 Am. Nat.). In response, we have modified the sentence to now read (lines 106-109 on our revised manuscript):

“The estimated (using the Β Function) maximum temperature at which spores were able to cause infections was 5 °C lower under fluctuating temperatures than under constant temperatures (Figure 2A; T_max_ = 25 °C for fluctuating vs. 30 °C for constant; confidence intervals for T_max_ do not overlap).”

3. Please make it easy to understand the word *exposure*… across the ms you expose Daphnia to different temps, but also to pathogens, and unexposed controls were exposed to a Placebo. Figure 3 legend should be more specific, if you want to expose the reader to some clarity!

Indeed, an excellent point. As infectious disease biologists, we often associate the word “exposure” with parasite exposure but, as the reviewer correctly points out, in our experiment we expose the animals to a wide variety of treatments. Accordingly, we have changed mentions of exposure throughout the manuscript to now specify explicitly what the animals were exposed to. That is; controls were exposed to a *placebo*, parasite treatments were exposed to *Ordospora* and these were crossed with exposure to different temperature regimes. We hope that this clarifies any unclarities and confusion in the text, figures, and legends.

Reviewer #2 (Recommendations for the authors):As I mentioned I have no major concerns about the manuscript, I liked it very much.My points (or advices) are as follows:– The first (lines 184-187) and the third (lines 188-190) sentences of the discussion basically deliver the same information. I'd suggest to remove the latter.

We agree that these sentences are indeed redundant and have removed the second sentence as suggested.

– lines 196-197: "Infection rates were reduced at higher temperatures when animals experienced diurnal fluctuations but not after experiencing a heatwave." – given that (as authors suggest in the Results section – lines 108-111) the authors lack the data over 22C for heat wave treatment (for absolutely justified reasons), the statement that infection rates were not reduced at higher temperatures in this treatment has a limited support. It is always risky to draw conclusions from data extrapolated beyond the last measurement point. Therefore, I think it would be good to add to the above mentioned statement a similar in character 'warning' as authors did in the Results section.

We agree fully. It would indeed not be recommended to extrapolate the results beyond the range of the data. Given that the heatwave at 22 ℃, which reaches 28 ℃ is already near the thermal optimum of the parasite, we do not, however, expect it to have had a significant impact our results. Nevertheless, in response to the comment of the reviewer, we have added the following sentence (lines 194-198 on our revised manuscript):

“Our estimates of maximum temperature for the heatwave, however, have broad confidence intervals, likely due to the lack of data at high temperatures, and extrapolations of our results beyond our highest heatwave temperature (which was 22 °C and reached 28 °C during the three-day heatwave which was close to the estimated maximum thermal tolerance of the parasite) should be interpreted with caution.”